# Design and Implementation of a Non-Common-View Axis Alignment System for Airborne Laser Communication

Chenghu Ke [1], Yuting Shu [2], Xizheng Ke [2,3,*], Meimiao Han [2] and Ruidong Chen [2]

1 School of Information Engineering, Xi'an University, Xi'an 710065, China; chenghuke@xawl.edu.cn
2 Faculty of Automation and Information Engineering, Xi'an University of Technology, Xi'an 710048, China; 2210321176@stu.xaut.edu.cn (Y.S.); kambibiyo@163.com (M.H.); 13772002569@163.com (R.C.)
3 Shaanxi Civil-Military Integration Key Laboratory of Intelligence Collaborative Networks, Xi'an 710126, China
* Correspondence: xzke@xaut.edu.cn

**Abstract:** This paper proposes a non-common-view axis alignment method for the alignment requirements of airborne laser communication systems. The system consists of a ground transmitting end and an airborne relay terminal. The ground transmitting end uses a camera and a pan-tilt for image tracking, while the airborne relay end uses a two-dimensional mirror to control the beam to achieve non-common-view axis alignment between the transmitting and receiving sides. The working principle and process of both the transmitter and receiver of the non-common-view axis alignment system for airborne laser communication were compared with traditional wireless optical alignment methods. The design process of the two-dimensional mirror used in this paper is introduced, the scanning trajectory of the two-dimensional mirror is simulated and analyzed according to the beam scanning principle, and the field experiment link is set up to carry out the airborne laser communication experiment. The experimental results show that when the link distance is 10 m, the tracking errors of the system in the azimuth and pitch directions are 19.02 µrad and 22.35 µrad respectively, and the amplitude of the electrical signal output by the signal detector is 84.0 mV; When the link distance is 20 m, the tracking errors of the system in the azimuth and pitch directions are 39.66 µrad and 33.94 µrad respectively, and the amplitude of the electrical signal output by the signal detector is 23.0 mV. Using this method, the alignment can be completed without data return, and the establishment of the reverse link can also be realized while the transmission link is quickly established, and there is no need for an air stability platform. The feasibility of the application of the non-common-view axis alignment method to the airborne laser communication system is verified.

**Keywords:** airborne laser communication; non-common-view axis alignment; acquisition; pointing; and tracking; two-dimensional mirror

## 1. Introduction

Wireless optical communication is a communication method that uses laser beams to carry signals to transmit in free space. It has the advantages of a high transmission rate and good security and can be applied to long-distance signal transmission, secure communication, and other fields [1,2]. Traditional aircraft-to-ground, aircraft-to-air, and relay communications mostly use radio frequency signals for transmission. With the development of wireless optical communication technology and related detection technologies, the application range of wireless optical communication technology is gradually expanding [3–5]. An airborne laser communication system is a wireless optical communication terminal carried on air platforms such as airships, drones, and aircraft. It is not only an important part of the construction of a wireless optical communication network, but also an important relay communication node in the communication link between satellite-ground and ground [6]. It is easily affected by the airborne platform's vibration and the atmospheric channel's random disturbance. The disturbance caused by this will cause the

beam's position to shift, and even the receiving end will lose the information on the spot, eventually leading to the interruption of the communication link [7]. Therefore, an accurate and fast acquisition, pointing, and tracking (APT) mechanism is needed to assist in the establishment of airborne laser communication links [8,9].

The APT system requires high precision and continuous alignment of the beam and can overcome the influence of mechanical vibration and the external environment on the system [10]. Wang Fuchao designed a full tracking controller applied to fast mirrors, using the full tracking method to effectively reduce the steady-state error of the system and expand the control bandwidth of fast mirrors [11]; Wang Junyao and others proposed a beam tracking method, which uses a rotating double prism for beam tracking, and uses a fast mirror to correct the optical axis deviation of the double prism for compound tracking [12]; Gao Lu proposed a two-dimensional optical phased array mirror structure based on the Gires-Tournois (optical standing wave resonator) resonator. The high-speed phase delay generated by the resonator is used to control the beam deflection angle. The simulation results show that the structure can achieve a deflection angle of 11.2° [13]; Antonello and others proposed a high-precision tracking and aiming system applied to satellite laser communication. Through indoor simulation experiments, the alignment error value of the system was measured to be better than 10 μrad [14]; Ke Xizheng proposed a method of alignment using a two-dimensional mirror as an actuator, and carried out far-field experiments of 1.3 km and 10.3 km. Experimental results show that this method can effectively reduce the tracking variance of the spot centroid [15].

This paper proposes a method to realize airborne laser communication by using a camera and a gimbal to track targets at the transmitter on the ground and using a two-dimensional mirror to control beam alignment at the airborne relay. The two-dimensional mirror is scanned and captured by changing the angle of the two-dimensional mirror in the pitch and azimuth directions, and the signal on the position sensor(PSD) is used as the feedback information of the relay terminal for tracking, without the need for an air stabilization platform.

## 2. System Working Principle

### 2.1. System Structure

The airborne laser communication system designed in this paper mainly comprises a signal transceiver system, an image tracking system, and a two-dimensional mirror control system. The overall structure is shown in Figure 1.

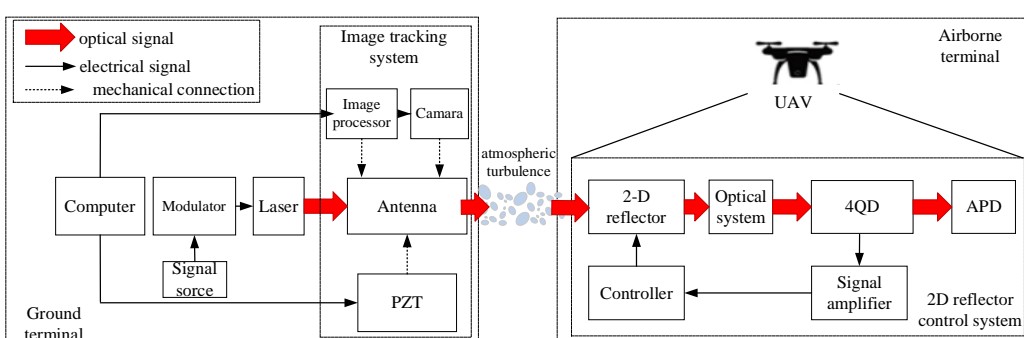

**Figure 1.** Structure diagram of airborne laser communication.

The signal transceiver system includes a signal source, laser tube, modulation driver, transmitting antenna, APD photodetector, etc. The signal to be transmitted is loaded on the laser by the modulation driver, and the signal light is collimated and emitted through the transmitting antenna; the ground-side image tracking system includes a tracking camera, a two-dimensional aiming platform, and a host computer. When the position and attitude of the UAV deviate, the deflection angle of the gimbal can be adjusted in real-time to control the emission direction of the beam, and realize long-distance coarse alignment from the ground-side transmitting antenna to the airborne two-dimensional

mirror; The two-dimensional mirror control system is located on the airborne side and is mainly composed of a stepping motor and a subdivision driver, an optical mirror, an embedded controller, a through-hole four-quadrant detector(4-QD), and a signal processor. When the light beam reaches the surface of the UAV, the acquisition, pointing, and tracking functions of the light beam are completed by driving the two-dimensional mirror, and the precise alignment between the two-dimensional mirror and the through-hole four-quadrant detector is realized.

### 2.2. Alignment Principle

Traditional wireless optical communication systems require the transmitter and receiver to be aligned on the same boresight, and the alignment principle is shown in Figure 2a. In the wireless optical communication link, the position information of the transmitting end and the receiving end is obtained utilizing the global position system (GPS) positioning system, and the position coordinates are converted into the servo execution amount of the APT system in the pitch direction and the azimuth direction, and then it is sent to the other side through the radio frequency signal for adjustment. Before controlling the alignment of the beam, the transmitting antenna needs to be controlled to scan due to the uncertain region between the transmitting antenna and the receiving antenna. When a light spot appears on the surface of the position detector (PSD) at the receiving end, the scanning is stopped, and the position information of the light spot is fed back to the transmitting end, and the transmitting end adjusts the deflection angle of the servo mechanism according to the received signal to control the beam to complete the alignment. When disturbed by atmospheric turbulence and mechanical vibration, the method of transmitting spot position information to the transmitter is still used to control the beam for tracking. It can be seen that, in the workflow of the traditional wireless optical APT system, the sending and receiving parties need to continuously transmit position signals. When the distance of the wireless optical communication link is large, it will be affected by atmospheric turbulence in the process of signal return, which is prone to signal delay, causing the sending and receiving parties to be unable to receive the spot position information at the current moment in time, and eventually lead to the interruption of the wireless optical communication link, greatly increasing the time and uncertainty required to establish the wireless optical communication link.

In the airborne laser communication system, due to the strong mobility of the flying platform, it is impossible to establish a wireless optical communication link using the traditional common-view axis alignment method. The method proposed in this paper for non-common-view axis alignment using a two-dimensional mirror is shown in Figure 2b. This method decomposes the traditional long-distance alignment of wireless optical communication into two parts of the alignment system, which are the coarse alignment between the tracking camera and the two-dimensional mirror at the airborne end and the fine alignment between the two-dimensional mirror and the four-quadrant detector [16]. The transmitter keeps the drone always in the center of the camera's field of view by driving the servo mechanism. At this time, after the laser is collimated and emitted by the transmitting antenna, it covers the surface of the airborne end, and the long-distance coarse alignment from the ground end to the airborne end is completed. When the airborne end receives the optical signal, the two-dimensional mirror starts to scan and capture the program. When the light spot information appears on the PSD device, the motor decelerates and adjusts until the light spot is located at the center of the position detector to complete the fine alignment. Finally, the light beam is focused on the surface of the photodetector(APD) behind the through-hole 4-QD in the form of a point light source, the six-pointed star pattern is used to simulate the position of point light source, and the light beam is sent back from the original optical path to the ground end for reception through electro-optical conversion. During the alignment process, since the two can be operated at a single end, it effectively solves the problem that the traditional alignment method needs to transmit the position information of the sender and receiver through radio

frequency signals, and the operation is simple and the complexity of the system is reduced. In addition, the two-dimensional mirror has a large dynamic adjustment range, which can quickly execute the scan capture procedure, saving capture time compared with traditional alignment methods.

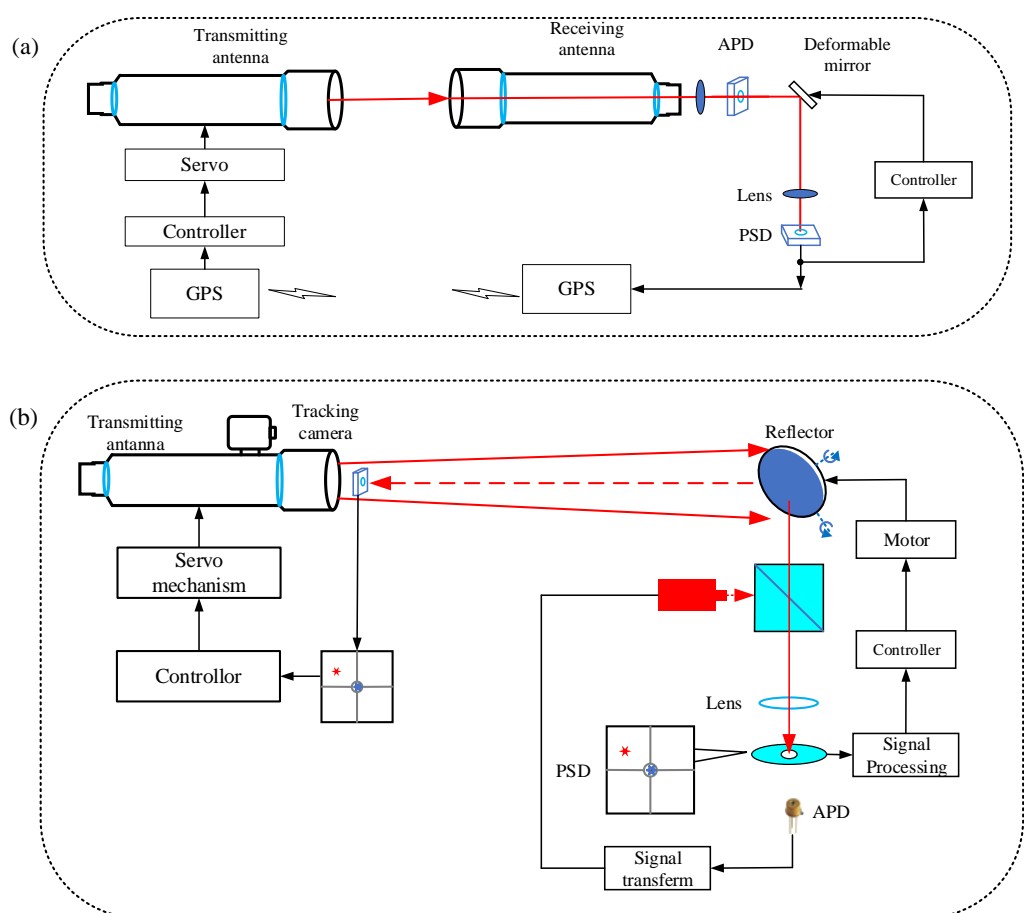

**Figure 2.** Beam alignment schematic diagram. (**a**) Common-view axis beam alignment; (**b**) Non-common-view axis beam alignment.

Figure 3 shows the workflow of the airborne laser communication non-common-view axis alignment system. Figure 3a,b are the workflows of the ground tracking system and the airborne receiver system, respectively. The ground transmitter uses the camera to extract the image features of the UAV and drives the gimbal to deflect according to the change of the image feature information of the UAV, to keep the image of the UAV in the center of the camera's field of view. After alignment, the laser can cover the surface of the UAV. The alignment process of the airborne end is the process of grating scanning by using a two-dimensional mirror until a light spot appears on the surface of the four-quadrant detector, and adjusting the motor angle to control the outgoing beam to pass through the central through hole of the four-quadrant detector.

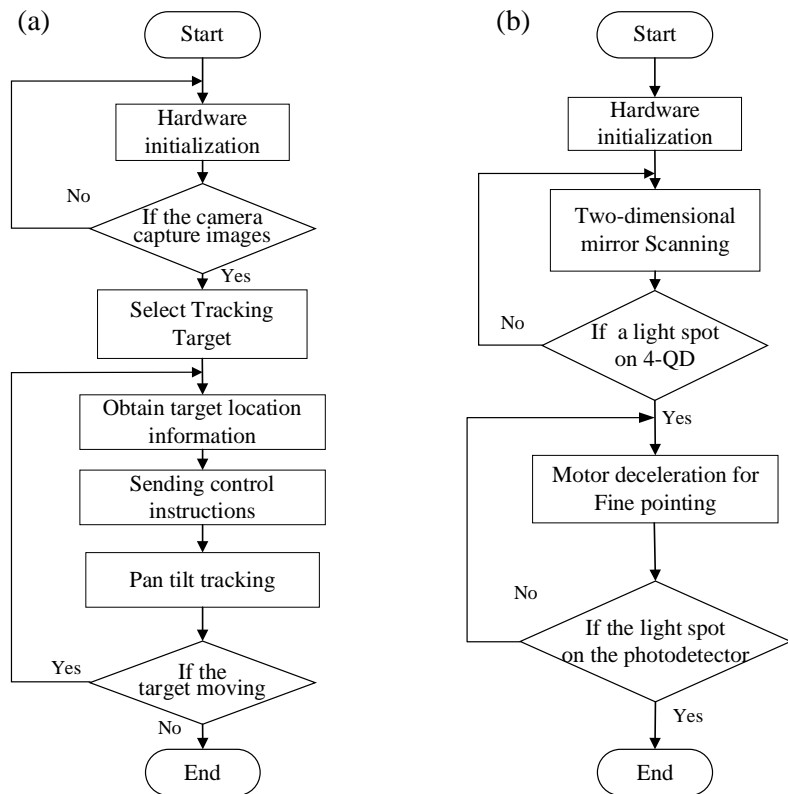

**Figure 3.** Flow chart of APT system for airborne laser communication. (**a**) Transmitting terminal; (**b**) Airborne terminal.

### 2.3. Working Principle of the APT System

In airborne laser communication systems, piezoelectric ceramic or electromagnetically driven mirrors are often used to achieve high-precision alignment and tracking, but their structures are complex and difficult to process, and special control boxes are required for driving. To realize the miniaturization and light weight of the airborne end alignment system, a two-dimensional mirror driven by a stepping motor is used as the main actuator of the airborne end in this paper. The structure of the two-dimensional mirror is shown in Figure 4, which consists of two two-phase hybrid stepping motors, an optical mirror, and an optical precision turntable. By controlling the rotation of the mirror in the two directions of pitch and azimuth, a large-scale and high-frequency dynamic adjustment can be realized, and the rotation of 0~360° and 0~160° can be realized in the pitch direction and the azimuth direction, respectively. At the same time, to make the angle of each step of the motor smaller, a subdivision driver is used to subdivide the basic step angle of the stepping motor to achieve higher alignment accuracy.

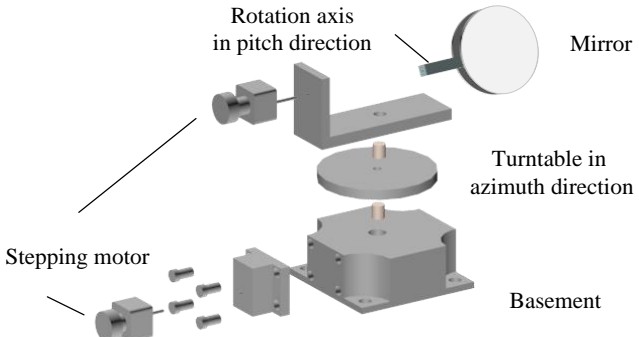

**Figure 4.** Structure diagram of a two-dimensional mirror.

After the emitted beam covers the surface of the airborne end, the scanning capture procedure is performed by the two-dimensional mirror. The two-dimensional mirror can realize the azimuth movement of $-90\sim90°$ and the pitch movement of $0\sim360°$, and the beam capture process is completed by raster scanning. The reflection principle diagram of the two-dimensional mirror beam is shown in Figure 5. $A$ is the incident light vector, $A'$ is the beam vector reflected by the mirror, $N$ is the normal vector of the mirror, $\theta_1$ and $\theta_2$ are the angles of deflection of the two-dimensional mirror around axes $C_1$ and $C_2$, respectively, and $\alpha$ is the incident angle.

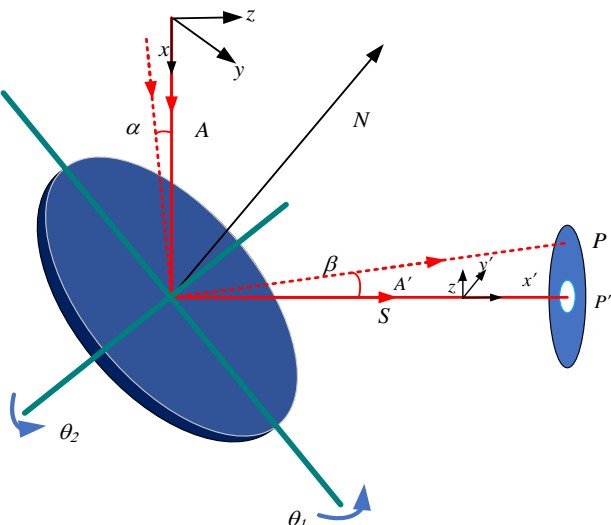

**Figure 5.** Schematic diagram of two-dimensional mirror beam reflection.

According to the reflection theorem of geometric optics, the outgoing ray $A'$ and the normal vector $N$ can be expressed as [17]:

$$A' = A - 2(A \cdot N)N \tag{1}$$

Let the components of $A$, $N$, and $A'$ in the $XYZ$ coordinate system be brought into Equation (1) to obtain:

$$\begin{cases} A'_x = (1 - 2N_x^2)A_x - 2N_xN_yA_y - 2N_xN_zA_z \\ A'_y = -2N_xN_yA_x + (1 - 2N_y^2)A_y - 2N_yN_zA_z \\ A'_z = -2N_xN_zA_x - 2N_yN_zA_y + (1 - 2N_z^2)A_z \end{cases} \tag{2}$$

According to Formula (2), reflection is a linear transformation, so Formula (1) can be written as follows:

$$A' = RA \tag{3}$$

$R$ is the reflection matrix, which can be expressed as:

$$R = \begin{bmatrix} 1 - 2N_x^2 & -2N_xN_y & -2N_xN_z \\ -2N_yN_x & 1 - 2N_y^2 & -2N_yN_z \\ -2N_zN_x & -2N_zN_y & 1 - 2N_z^2 \end{bmatrix} \tag{4}$$

In Formula (4), $N_x$, $N_y$, and $N_z$ are the projections of the normal vector of the mirror in its reference coordinates. Assuming that the distance between the reflector and the four-quadrant detector is $S$, the coordinates of the spot on the plane of the four-quadrant detector can be expressed as [15]:

$$\begin{cases} D_x = \lambda_x \dfrac{(V_A + V_B) - (V_C + V_D)}{V_A + V_B + V_C + V_D} = \lambda_x \dfrac{(I_A + I_B) - (I_C + I_D)}{I_A + I_B + I_C + I_D} = \lambda_x \dfrac{(S_A + S_B) - (S_C + S_D)}{S_A + S_B + S_C + S_D} \\ D_y = \lambda_y \dfrac{(V_A + V_C) - (V_B + V_D)}{V_A + V_B + V_C + V_D} = \lambda_y \dfrac{(I_A + I_C) - (I_B + I_D)}{I_A + I_B + I_C + I_D} = \lambda_y \dfrac{(S_A + S_C) - (S_B + S_D)}{S_A + S_B + S_C + S_D} \end{cases} \tag{5}$$

According to the Formula (5), the scanning trajectory of the two-dimensional mirror is obtained by simulation, as shown in Figure 6:

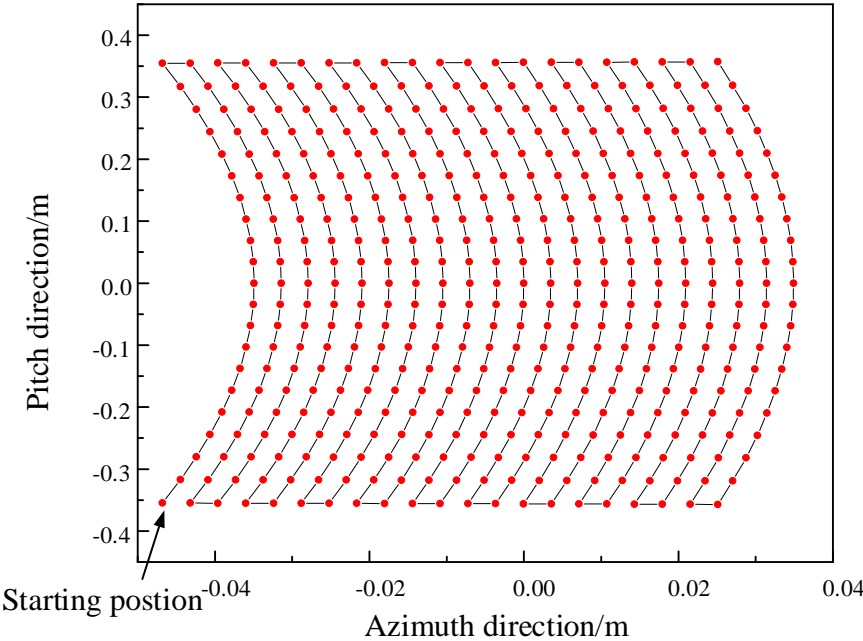

**Figure 6.** Scanning trajectory of the beam emitted from a two-dimensional mirror.

The system uses the position of the light spot on the four-quadrant detector as feedback information to adjust the pitch angle and azimuth angle of the two-dimensional mirror. When the light spot reaches different quadrants, the position information of the light spot is calculated according to the voltage value of each quadrant on the surface of the detector, that is, the offset of the light spot in the *x-y* direction $D_x$, $D_y$, and the calculation formula of the light spot position is shown in Formula (6) [18]:

$$\begin{cases} D_x = \lambda_x \dfrac{(V_A + V_B) - (V_C + V_D)}{V_A + V_B + V_C + V_D} \\ D_y = \lambda_y \dfrac{(V_A + V_C) - (V_B + V_D)}{V_A + V_B + V_C + V_D} \end{cases} \tag{6}$$

In the formula, $\lambda$ is a proportional coefficient, and its size is related to the energy distribution and size of the spot. Its function is to convert the value in Formula (6) into the distance between the spot and the center of the four-quadrant detector, and $V$ is the voltage value of each quadrant. After obtaining the position information of the spot, calculate the offset of the spot from the center of the four-quadrant detector as $\Delta D_x$, $\Delta D_y$, and calculate the deflection angle of the beam, as shown in Formula (7) [19,20]:

$$\begin{cases} \theta = \lambda_x \cdot \arctan \dfrac{\sqrt{\Delta D_x^2 + \Delta D_y^2}}{d} \\ \phi = \lambda_y \cdot \arctan \dfrac{\Delta D_x}{\Delta D_y} \end{cases} \tag{7}$$

In the formula, $\theta$ the pitch angle $\phi$ is the azimuth angle, and $d$ is the distance from the receiving surface to the photosensitive surface of the detector. This formula expresses the corresponding relationship between the experimental measurement value and the actual value.

Assuming that the error angles corresponding to the azimuth direction and the pitch direction are $x_i$ and $y_i$ respectively, for $n$ measurements, the radial angle deviation $r_i$ and its average value $\bar{r}$ can be expressed as:

$$r_i = \sqrt{x_i^2 + y_i^2}, \ \bar{r} = \frac{1}{n}\sum_{i=1}^{n} r_i \tag{8}$$

Then the $3\sigma$ alignment accuracy can be expressed as:

$$\varepsilon = \bar{r} + 3\sqrt{\frac{1}{n-1}\sum_{i=1}^{n}(r_i - \bar{r})^2} \tag{9}$$

## 3. Methods

### 3.1. System Composition

Figure 7 shows the structure diagram of the airborne laser communication experiment. The ground transmitting end is composed of a 650 nm laser, signal modulator, transmitting antenna, tracking camera, PC, and servo pan-tilt; the relay platform is composed of two-dimensional mirror, motor and its driver, position sensor, photodetector, and single-chip microcomputer controller. After the coarse alignment from the transmitting antenna to the two-dimensional mirror, the camera is used to select the image of the UAV, and the pitch angle and azimuth angle of the pan-tilt are adjusted according to the moving position information of the UAV so that the beam completely covers the relay system surface. The two-dimensional mirror executes the scan capture program until there is a light spot on the surface of the four-quadrant detector, and then the motor slows down for fine alignment so that the beam passes through a hole in the center of the four-quadrant detector surface. At this time, the light beam reaches the surface of the photodetector, and the signal light is loaded to a new laser through the electrical-optical conversion circuit, and the optical signal is reflected to the ground receiving end through the beam splitter prism and the two-dimensional mirror. Table 1 shows the main equipment and parameters of the experiment.

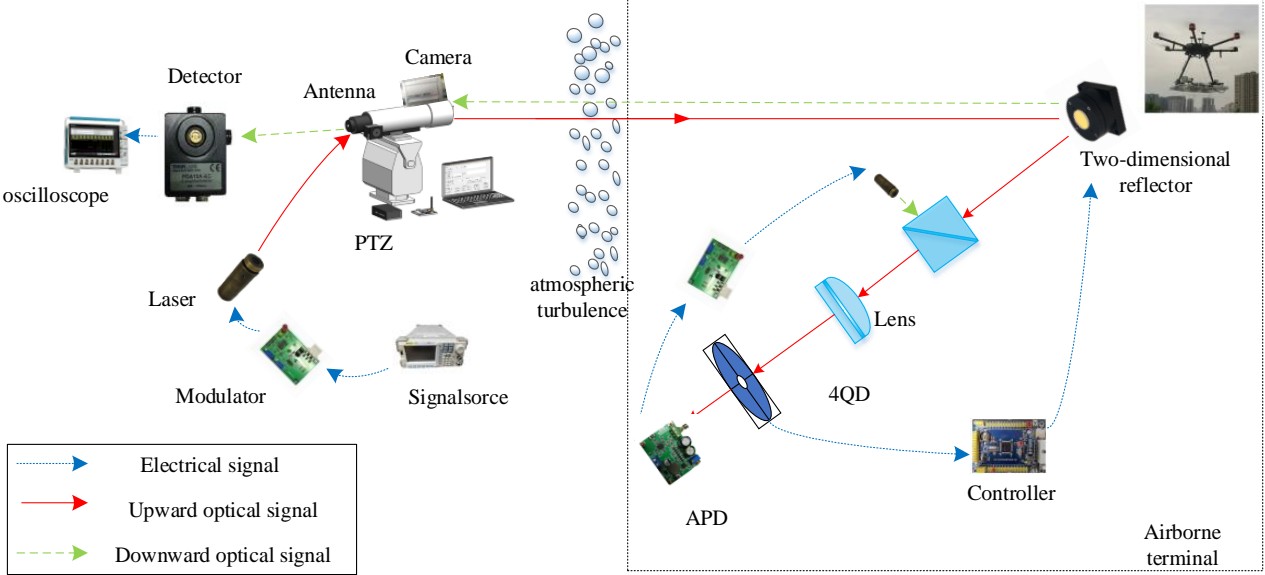

**Figure 7.** Structure diagram of airborne laser communication experiment.

**Table 1.** Experimental equipment and parameters.

| Equipment | Parameters |
|---|---|
| Laser | wavelength: 650 nm<br>power: 80 mW |
| Tracking camera | zoom factor: 128<br>pixel size: 1 pixel = 20 μm |
| Two-dimensional aiming gimbal | adjustment range: 0~360° (orientation)<br>−70–70° (pitch) |
| | maximum load weight: 10 kg |
| Piezoelectric micro-motion gimbal | model: PT2 K |
| | loading capacity: 4 kg |
| | voltage input range: 0~10 V |
| | driving mode: piezoelectric ceramics |
| | resolution:0.01 μrad |
| Antenna | aperture: 105 mm |
| Two-dimensional mirror | mirror surface: 60 mm<br>Adjustment range: −90~90° (orientation)<br>0–360° (pitch) |
| Photodetector | type: InGaAs<br>cut-off frequency: 30 kHz~1.5 GHz |
| Four-quadrant detector | photosensitive surface diameter: 5.05 mm<br>response time: 13 ns |

Figure 8 is a schematic diagram of the airborne laser communication field experiment. The ground transmitting end and the airborne end are located on the north side and the center of the Xi'an University of Technology playground, respectively. The experiment time is 29 May 2023. The weather is fine, the temperature is 18 °C, and the northeast wind is 3–4.

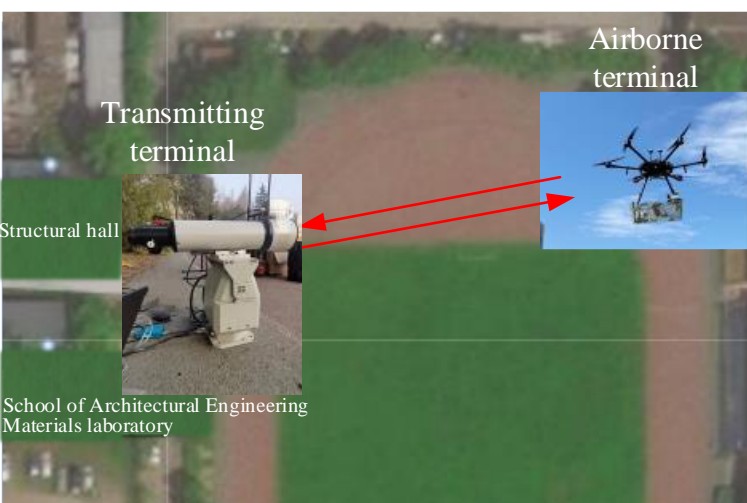

**Figure 8.** Schematic diagram of airborne laser Communication field experiment.

### 3.2. Ground End Alignment Experiment

Figure 9 is a schematic diagram of calibration and coarse alignment using the camera and the pan-tilt, and the image position of the drone is used as the feedback information for adjusting the gimbal rotation. Firstly, the upper computer interface selects the UAV and the airborne terminal as calibration targets. By adjusting the deflection angle of the pan-tilt and the zoom factor of the camera, the ground pan-tilt is driven to move to keep

the UAV in the center of the camera field of view, and the long-axis coarse alignment between the transmitting antenna and the two-dimensional mirror of the relay terminal is completed. After the alignment, the laser beam can cover the surfaces of the UAV and the relay terminal.

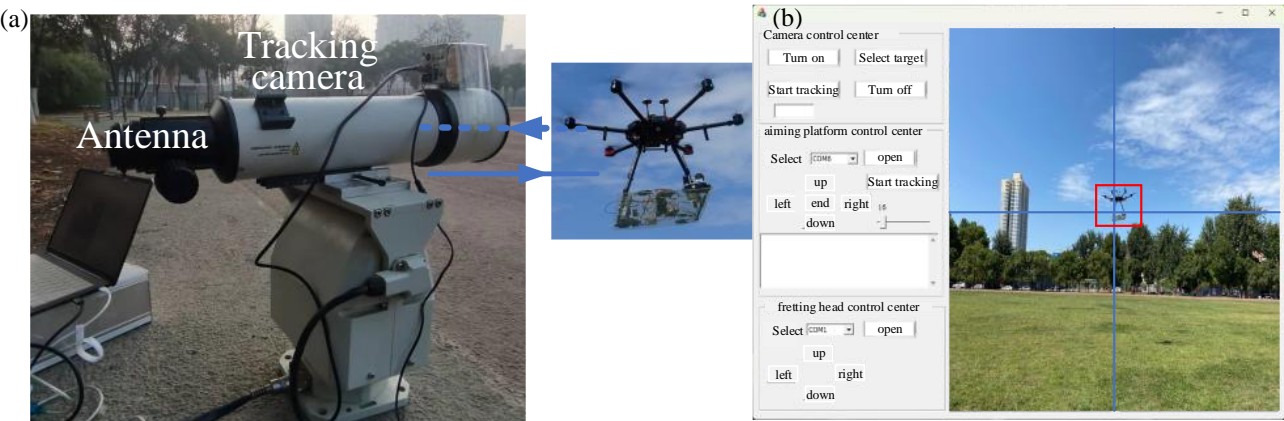

**Figure 9.** Schematic diagram of ground terminal alignment. (**a**) Transmitting terminal; (**b**) Upper computer interface.

After the coarse alignment is completed, tracking is performed by calculating the pixel point deviation value between the center of mass position of the drone and the center point of the camera's field of view. As shown in Figure 10a,b, the tracking curves in the pitch direction and azimuth direction of UAV during mobile flight are shown respectively. The longitudinal axis represents the size of pixels, and the number of breakpoints is the number of times pan-tilt adjustment. As can be seen from the figure, the azimuth was adjusted 9 times, and the pitch angle was adjusted 4 times.

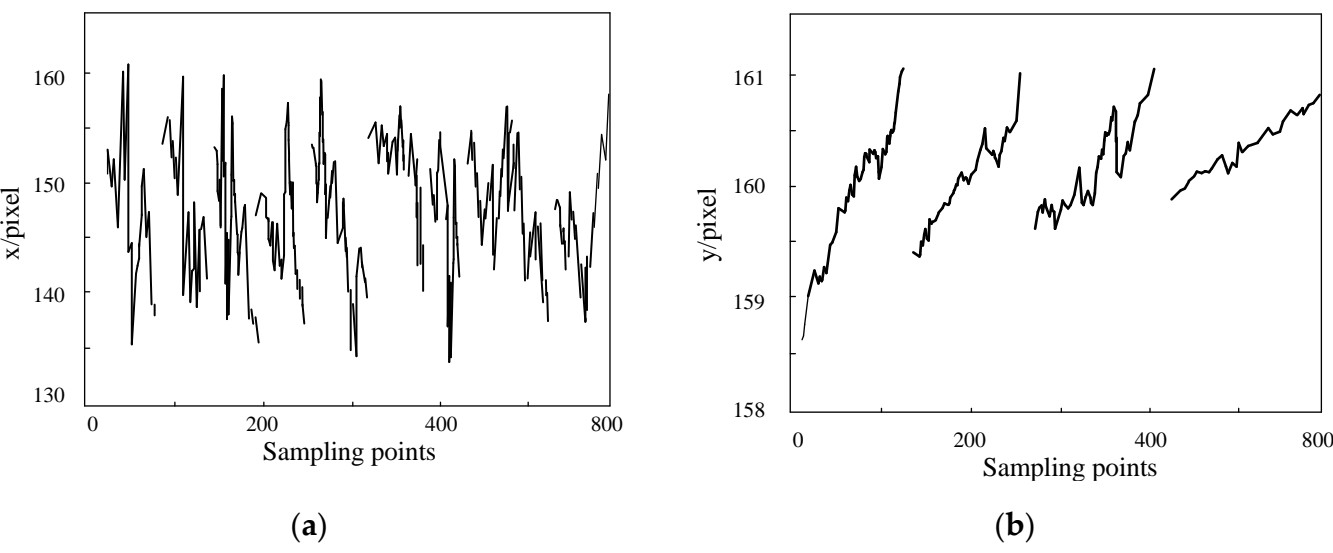

**Figure 10.** Ground terminal tracking curve. (**a**) Azimuth direction; (**b**) Pitch direction.

### 3.3. Airborne End Fine Alignment Experiment

As shown in Figure 11, it is the change of the output voltage of each quadrant of the four-quadrant detector during the flight state of the UAV during take-off, movement, and hovering. It can be seen from the figure that at the 0 sampling point, the voltage values of the first and fourth quadrants are the highest, that is, after the capture is completed, the light spots fall on the first and fourth quadrants. At sampling points 50–200, after the motor decelerates and executes the fine alignment command, the voltage changes in each

quadrant are relatively gentle. During the flight, the motor continuously adjusts the pitch angle and azimuth angle to complete the tracking, and the voltage value of each quadrant changes greatly at this stage; after the sampling point 3800, the UAV is in the hovering stage, and the voltage value of each quadrant changes gradually.

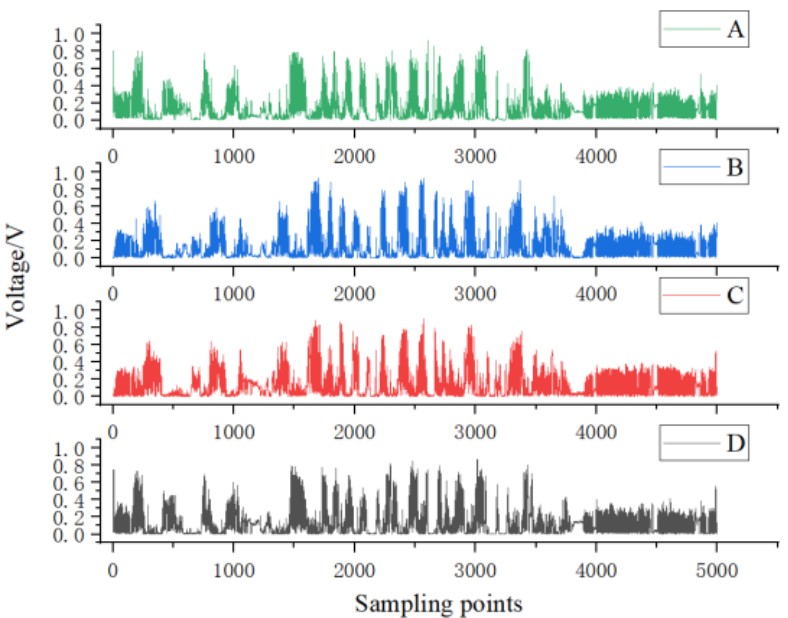

**Figure 11.** Four-quadrant detector voltage variation curve for each quadrant.

Figure 12 shows the variation of the voltage difference in the pitch and azimuth direction of the through-hole four-quadrant detector, in which Figure 12a shows the voltage difference variation curve in the azimuth direction, and Figure 12b shows the voltage difference variation curve in the pitching direction. It can be seen from the figure that the voltage difference is too large in the range of sampling points 0–3700. At this time, the UAV is in the mobile flight stage, and the adjustment range of the two-dimensional mirror is relatively large; after the sampling point of 3700, the voltage difference is close to 0. At this time The UAV is in a hovering flight state, the adjustment range of the two-dimensional mirror is small, and the voltage difference curve gradually tends to be stable.

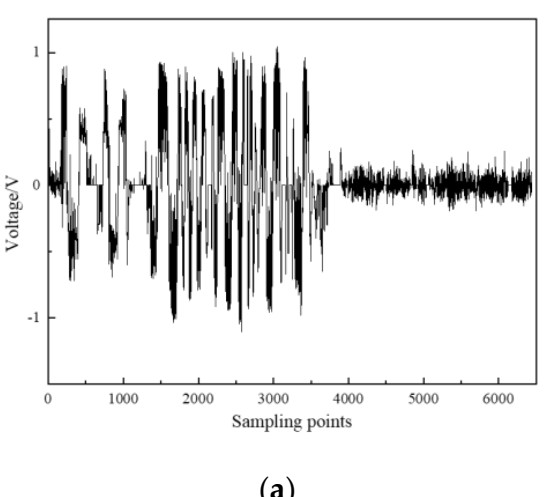

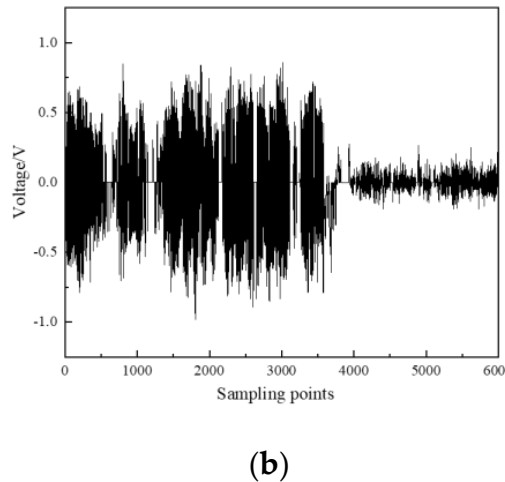

**(a)**          **(b)**

**Figure 12.** Four-quadrant detector x-y axis voltage change curve. (**a**) X-axis direction; (**b**) Y-axis direction.

As shown in Figure 13, the spot position distribution calculated according to the voltage difference on the four-quadrant detector has a total of 6300 data points. As shown in Figure 14, since there is a hole in the center of the four-quadrant detector, there is no voltage data at the hole after alignment, so there is a hole in the middle of the spot position distribution, and the closer to the central through hole, the higher the alignment degree. After calculating the distance from the coordinates of each spot to the center of the detector, the pitch angle and azimuth angle errors of the two-dimensional mirror are obtained through calculation, and the tracking error of the system is calculated to be 13.98 μrad(3σ) according to Formula (9).

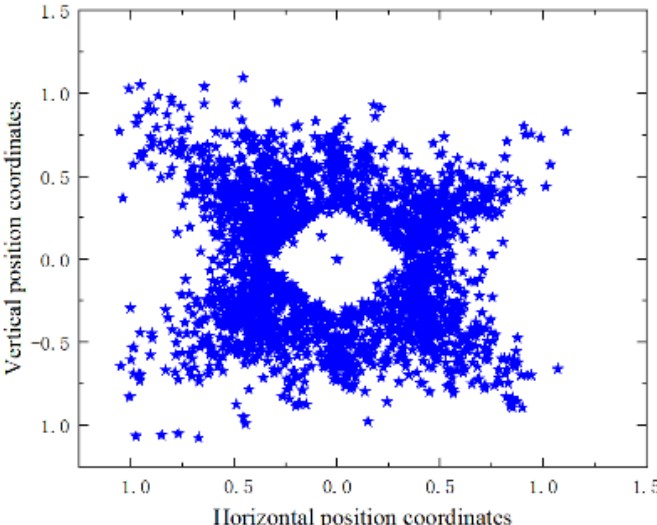

**Figure 13.** Spot position distribution diagram.

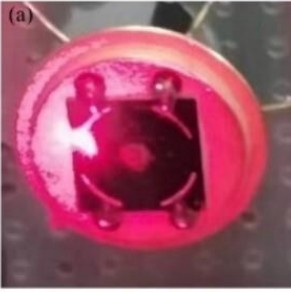
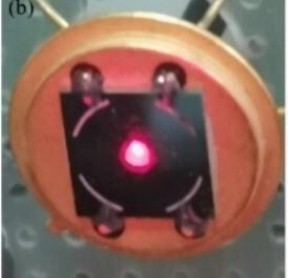

**Figure 14.** Align the position of the front and rear light spots (**a**) Before alignment; (**b**) After alignment.

When the communication link distance is 10 m, the light beam is captured 10 times by using the two-dimensional mirror on the airborne end, all of which are captured successfully and the average capture time is 31.2 s. In the tracking process, the tracking errors of the spot in the azimuth direction and the elevation direction of the through-hole four-quadrant detector are calculated respectively. From the test results in Figure 15, it can be seen that the tracking mean square error of the system in the azimuth direction is 19.02 μrad(3σ), and the tracking mean square error in the pitch direction is 22.35 μrad(3σ).

When the communication link distance is different, the signal optical power received by the airborne end is also different. As shown in Figure 16, when the communication link distance is 10 m, the optical power change curve is measured by the APD at the airborne end. It can be seen from the figure that the detected average optical power is about 6.05 μW, and the variance of the logarithmic amplitude fluctuation of the received power is $7 \times 10^{-7}$.

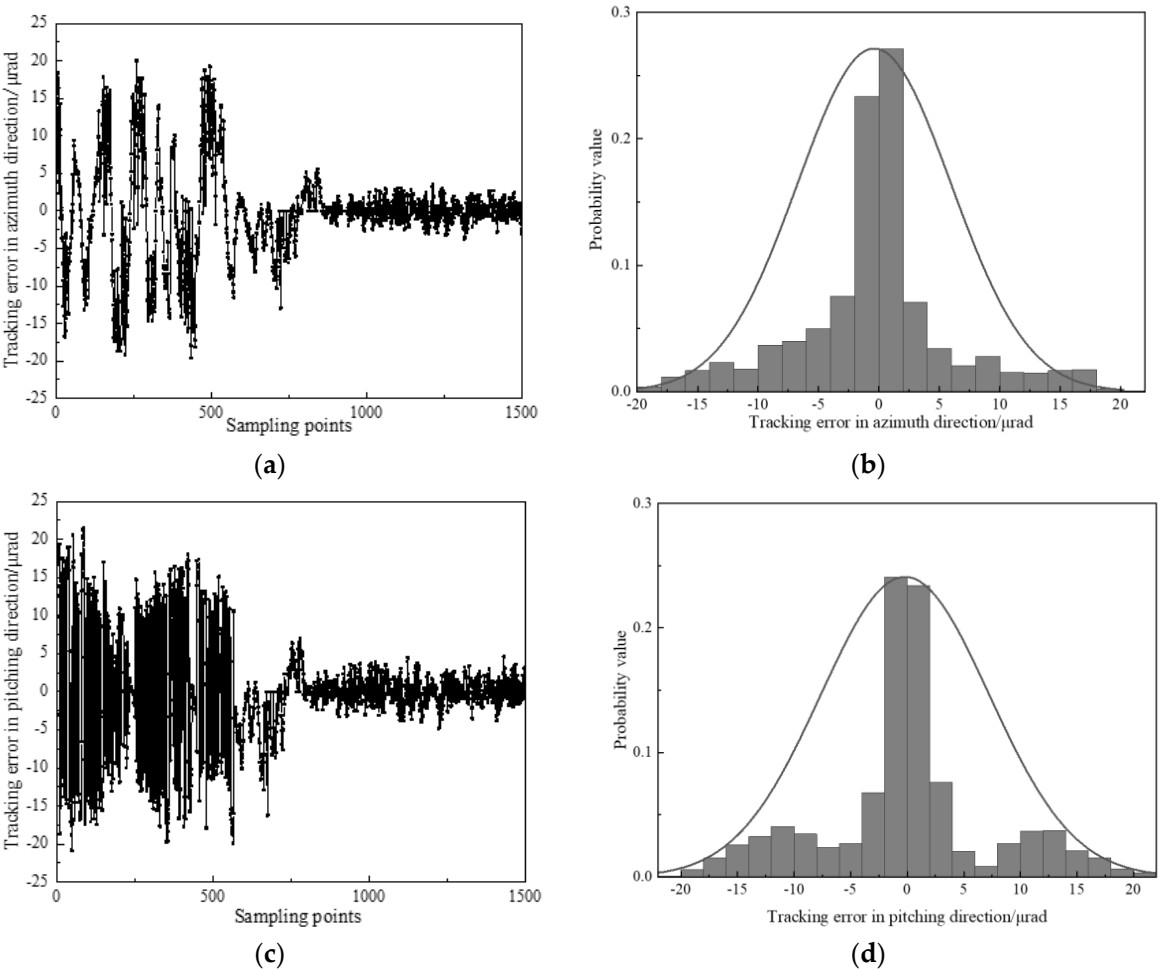

**Figure 15.** Tracking error curve of 10 m field experiment. (**a**) Tracking error in azimuth direction; (**b**) Statistical value of tracking error in azimuth direction; (**c**) Tracking error in pitch direction; (**d**) Statistical value of tracking error in pitch direction.

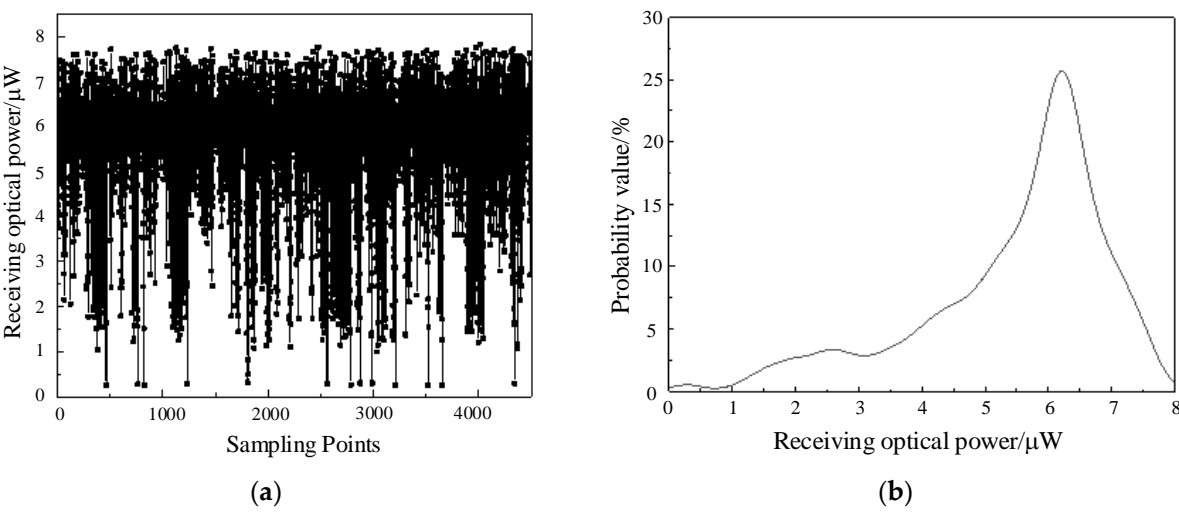

**Figure 16.** The change curve and distribution of received optical power at the airborne end of the 10 m field experiment. (**a**) The change curve of optical power; (**b**) The statistical value of optical power.

When the communication link distance is 10 m, the waveform of the transmitted signal and the waveform output by the detector at the ground end are shown in Figure 17a,b. The

laser power at the transmitting end is 80 mW, the signal amplitude is 300 mV, the signal frequency is 5 MHz/s, and the waveform amplitude output by the detector is 84.0 mV.

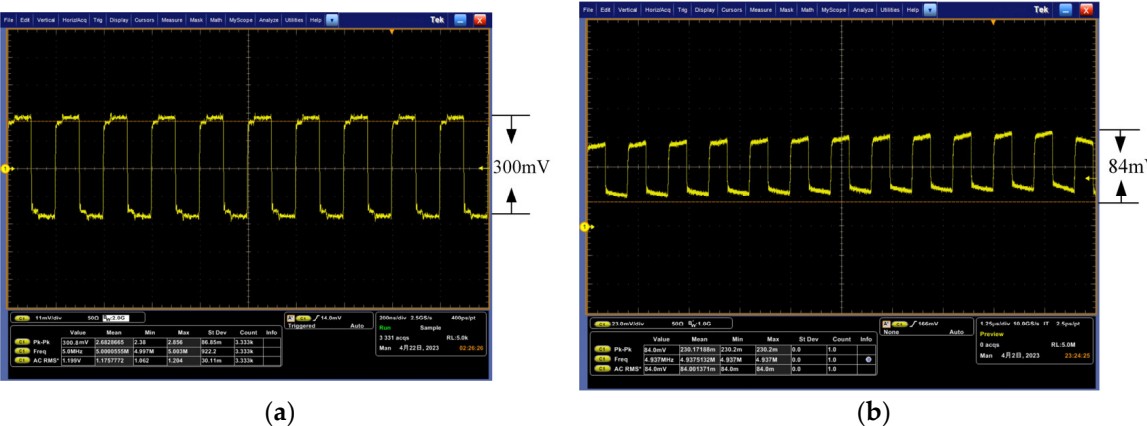

(**a**)                                                   (**b**)

**Figure 17.** Output waveform of the detector at a distance of 10 m. (**a**) Transmission signal; (**b**) Detection signal.

When the communication link distance is 20 m, the tracking errors of the spot in the azimuth and elevation directions of the through-hole four-quadrant detector are calculated respectively. From the test results in Figure 18, it can be seen that the tracking mean square error of the system in the azimuth direction is 39.66 μrad($3\sigma$), and the tracking mean square error in the pitch direction is 33.94 μrad($3\sigma$).

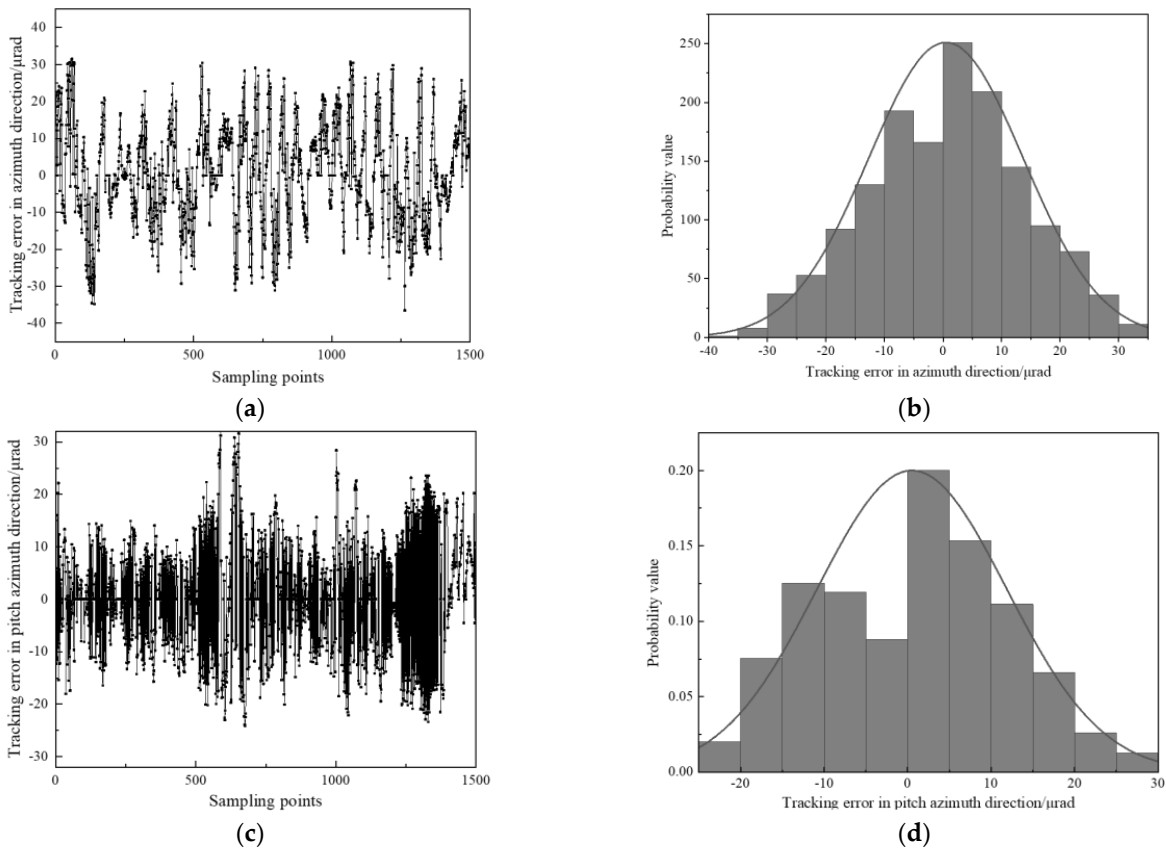

**Figure 18.** Tracking error curve of 20 m field experiment. (**a**) Tracking error in azimuth direction; (**b**) Statistical value of tracking error in azimuth direction; (**c**) Tracking error in pitch direction; (**d**) Statistical value of tracking error in pitch direction.

When the communication link distance is 20 m, the optical power change curve measured by the APD at the airborne end is shown in Figure 19. It can be seen from the figure that the detected average optical power is about 2.63 μW, and the variance of the logarithmic amplitude fluctuation of the received power is $8.04 \times 10^{-13}$.

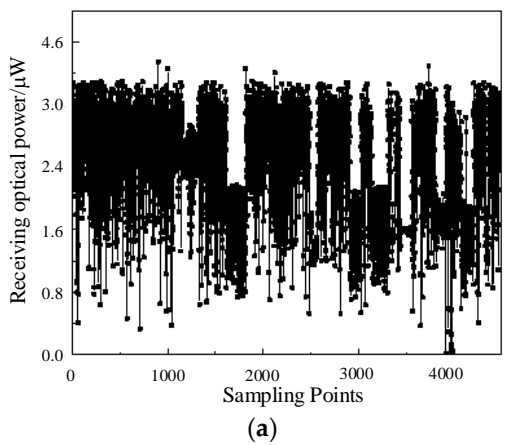 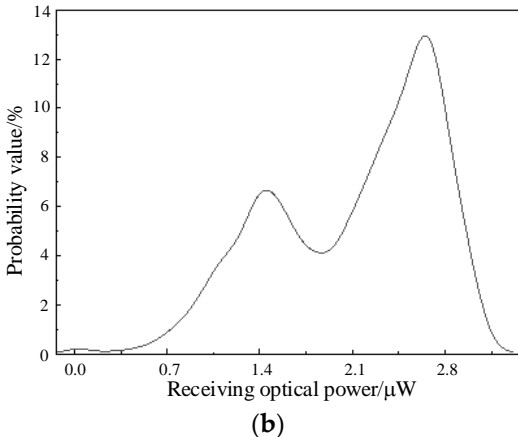

(**a**)            (**b**)

**Figure 19.** The change curve and distribution of received optical power at the airborne end of the 20 m field experiment. (**a**) The change curve of optical power; (**b**) The statistical value of optical power.

The waveform of the transmitting signal and the waveform output by the detector on the ground are shown in Figure 20a,b. The power of the laser at the transmitting end is 80 mW, the signal amplitude is 300 mV, the signal frequency is 5 MHz/s, and the communication link distance is 20 m, the waveform amplitude of the detector output is 23.0 mV.

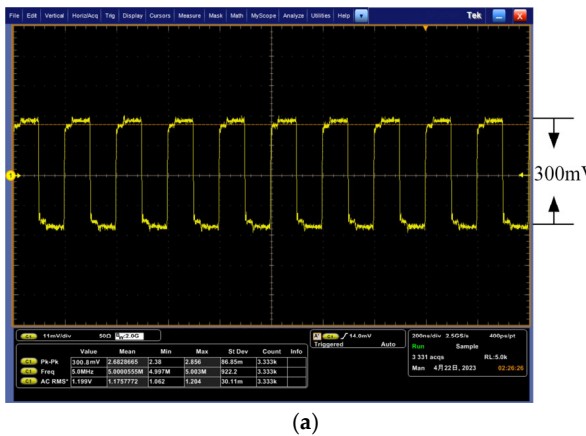 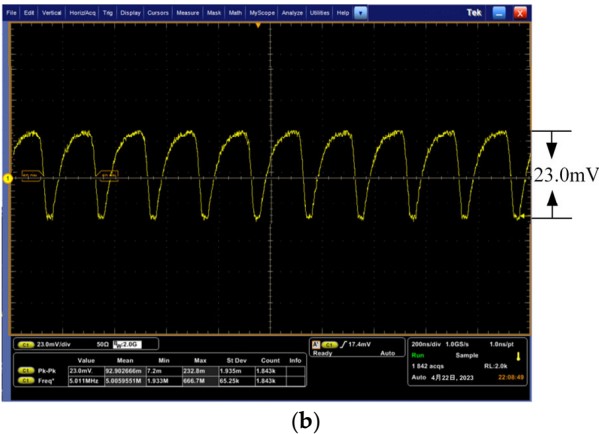

(**a**)            (**b**)

**Figure 20.** Output waveform of the detector at a distance of 20 m. (**a**) Transmission signal; (**b**) Detection signal.

In non-common-view axis communication, light does not propagate along an ideal straight-line path, but undergoes factors such as refraction and scattering. Therefore, as the distance of the communication link increases, when the signal reaches the receiving end, it will be affected by factors such as the external environment and optical devices, and part of the energy will be lost during transmission, so the received optical power will decrease, resulting in a decrease in the quality of airborne laser communication and an increase in the tracking error of the system.

*3.4. Discussion*

The traditional wireless optical communication system requires the transmitter to keep the same line of sight alignment with the receiver. When the distance of the communication link is large, it will produce signal delay and increase time and uncertainty. The method proposed in this paper changes the traditional joint control of the transmitter and the airborne side to achieve alignment and transforms it into a situation where both the transmitter and receiver can be independently controlled to achieve alignment, which is more flexible. When testing the capture performance, alignment performance, and spot positioning performance of the airborne system, it is found that, compared with the traditional method of common-view axis, it was greatly improved, and the problem of long time-consuming beam alignment in traditional wireless optical communication was solved. Non-common-view axis alignment system is an important technology used in airborne laser communication, which allows a certain angle deviation between transmitter and receiver in the process of optical communication. This is especially suitable for UAVs, aircraft, and other reasons that may cause orientation changes due to motion, weather, and other reasons. Based on the researIh in this paper, there are still some problems to be further solved and perfected. Numerical simulation should be carried out to analyze the maximum communication link distance under this system, which is helpful in analyzing the effect of airborne communication under larger distances. This experiment only realized the transmission of square wave signals. In the next step, the transmission of voice, video, and other signals can be realized by adding cameras, encoders/decoders, and other equipment, and the applicability will be strengthened and applicable to more fields. And the test environment is carried out under weak turbulent conditions, without considering the impact on the airborne laser communication system under strong turbulent weather conditions. In the later stage, the impact on the APT system can also be studied by comparing the experimental results under different weather conditions, and more stable communication can be achieved through technologies and algorithms such as adaptive optics, wavefront distortion correction, and anti-interference design.

**4. Conclusions**

In this paper, a non-common-view axis alignment system is designed to meet the alignment requirements of airborne laser communication. Among them, the coarse alignment of the ground transmitter can ensure that the transmitted beam covers the drone, and the airborne terminal can control the beam to achieve alignment by adjusting the pitch angle and azimuth angle of the two-dimensional mirror, and the sending and receiving parties do not need to transmit data back. Field experiments were carried out at link distances of 10 m and 20 m respectively, the experimental results show that the tracking accuracy of the system is 13.98 μrad, the signal amplitude at the receiving end is 23.0 mV, and the relative error of spot position detection is 0.97%, which is 36.6% higher than the traditional positioning algorithm. The feasibility of applying the non-common-view axis alignment method to the airborne laser communication system was verified.

**Author Contributions:** Conceptualization, X.K. and R.C.; methodology, X.K., M.H., R.C. and Y.S.; software, R.C. and M.H.; validation, C.K., Y.S. and X.K.; formal analysis, C.K.; investigation, Y.S. and R.C.; resources, C.K. and Y.S.; data curation, C.K. and Y.S.; writing—original draft preparation, C.K. and Y.S.; writing—review and editing, C.K., R.C. and Y.S. All authors have read and agreed to the published version of the manuscript.

**Funding:** Funding was received from the following: The Key Industrial Innovation Chain Project of Shaanxi Province [grant number 2017ZDCXL-GY-06-01]; the General Project of National Natural Science Foundation of China [grant number 61377080]; the Xi'an Science and Technology Plan (22GXFW0115); and the Scientific Research Team of Xi'an University (D202309).

**Institutional Review Board Statement:** The study did not require ethical approval.

**Informed Consent Statement:** The study did not require ethical approval.

**Data Availability Statement:** The data that support the findings of this study are available from the corresponding author upon reasonable request.

**Conflicts of Interest:** The authors declare no conflict of interest.

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
