# Peer review of "Design and Implementation of a Non-Common-View Axis Alignment System for Airborne Laser Communication"

_photonics, doi:10.3390/photonics10091037_

Round 1

Reviewer 1 Report

The authors disclose a non-common optical axis alignment system, designed for meeting the alignment prerequisites of airborne laser communication (ALC). The method is based on two independent alignment steps: a coarse alignment at the ground-based transmitter that ensures that the transmitted beam effectively covers the drone. At the aerial terminal, finer adjustment is achieved by tuning the pitch and azimuth angles of a two-dimensional mirror. Importantly, this adjustment process occurs without the need for data transmission between the transmitting and receiving parties, and closing a feedback loop is not necessary.

The paper is clearly written and well-illustrated by graphical schemes and pictures. I suggest the addition of 2 missing items:

1.        A short discussion on the applicability of the method and its limitations in terms of range and SNR, beyond the practically indoor conditions demonstrated.

2.       Technical details of critical components, e.g. gimbal and camera, model specs: resolution and speed.

Reviewer 2 Report

This paper designs an APT system applied to the UAV platform. Among them, the ground transmitter uses a camera and a gimbal to track the image of the UAV, and the airborne side uses a two-dimensional mirror to control the beam to achieve non-common-view axis alignment between the transmitter and receiver. The proposed method is innovative and has application prospects, and it is a hot issue in the field of airborne laser communication.

1. The ground transmitter uses cameras and gimbals for image tracking. What algorithm is used for image tracking and what is its principle? How to ensure the effectiveness of tracking?

2. The airborne end uses a two-dimensional mirror to control the beam to achieve non-common-view axis alignment between the transmitting and receiving sides. Please discuss/clarify how this problem is solved. What method is used? What steps are taken to ensure its accuracy?

3. Please explain the advantages and disadvantages of the proposed method now, and explain in detail how to improve it in future work.

Reviewer 3 Report

The manuscript entitled “Design and Implementation of a Non-common-view Axis Alignment System for Airborne Laser Communication” proposes an applied optical design technique for the performance improvement and structure simplification of airborne laser communication. This paper can be published in Photonics after a minor modification.

1.       In Fig. 7, the electrical and upward optical signals use the same marker, which is not conducive to distinguishing and reading, so I suggest making some modifications.

2.       More details about the 650 nm laser should be provided.

3.       The authors should elaborate in more detail on the contribution of this work to the performance improvement of the airborne laser communication system and complement the subsequent direction.

The quality of the English expression is moderate, and you can make some appropriate modifications.

Reviewer 4 Report

1. Numerical simulations should be carried out to analyze the maximal 

communication link distance.

2. As the communication link distance increases, the error increases. The authors should explain the reason.

3. Alignment principle is hard to understand. Many elements, such as APD, PSD, and deformable mirror in Fig. 2, have not been described.

4. Those Equations in 2.3 have not been used in the following part. The authors should use them in analyzing the following results.

5. Most of the references are in Chinese. The authors should cite more relevant work in English.

6. The Chinese annotations in Fig. 8 should be replaced by English annotations.

Reviewer 5 Report

The authors proposed a non-common-view axis alignment method for airborne laser communication. The manuscript is well written and the performance of the alignment method is verified by experiment. In my opinion, the manuscript can be accepted after major revisions. Some comments are listed as below.

1.     Detailed comparison with the traditional common-view axis alignment method should be added to better show the advantages of the proposed method.

2.     The environmental parameters may influence the accuracy of the alignment, such as the wind, rain, etc. Please discuss the impact of these environmental parameters on the system performance.

3.     The image of the drone is used to adjust the gimbal. However, when the distance between the transmitter and receiver is long, the image may be not acquired. Please discuss the maximum distance support by the proposed system.

4.     The authors gave the tracking errors of the system but did not analyze the origin of the tracking errors. Please analyze the reason why the tracking errors are existed and talk about the method to reduce the tracking errors.

5.     Please check the whole paper carefully. For example, in the caption of Figure 20, the distance should be 20 m.

Minor editing of English language required

Round 2

Reviewer 5 Report

The authors have clarified my concerns, and I have no additional comment for this revised manuscript.